# Comprehensive Cytokine Profiling of Patients with COVID-19 Receiving Tocilizumab Therapy

**DOI:** 10.3390/ijms23147937

**Published:** 2022-07-19

**Authors:** Anna Lebedeva, Ivan Molodtsov, Alexandra Anisimova, Anastasia Berestovskaya, Oleg Dukhin, Antonina Elizarova, Wendy Fitzgerald, Darya Fomina, Kseniya Glebova, Oxana Ivanova, Anna Kalinskaya, Anastasia Lebedeva, Maryana Lysenko, Elena Maryukhnich, Elena Misyurina, Denis Protsenko, Alexander Rosin, Olga Sapozhnikova, Denis Sokorev, Alexander Shpektor, Daria Vorobyeva, Elena Vasilieva, Leonid Margolis

**Affiliations:** 1Laboratory of Atherothrombosis, A.I. Yevdokimov Moscow State University of Medicine and Dentistry, 20 Delegatskaya Str., 127473 Moscow, Russia; angeoksana@gmail.com (O.I.); minutka86@mail.ru (A.K.); maryukhnich@gmail.com (E.M.); vorobyeva-daria2015@yandex.ru (D.V.); 2Clinical City Hospital Named after I.V. Davidovsky, Moscow Department of Healthcare, 11/6 Yauzskaya Str., 109240 Moscow, Russia; ivan.molodtcov@gmail.com (I.M.); anisimovaalexandra5@gmail.com (A.A.); dukhin.o@yandex.ru (O.D.); ton969@yandex.ru (A.E.); kseniya941996@gmail.com (K.G.); lebedevaay@zdrav.mos.ru (A.L.); gkb23@zdrav.mos.ru (A.R.); olg.sapozhnikowa@yandex.ru (O.S.); sokorevda@gmail.com (D.S.); moscowcardio_23@yahoo.com (A.S.); 3Clinical City Hospital №40, Moscow Department of Healthcare, 7 Kasatkina Str., 129301 Moscow, Russia; mmcc@zdrav.mos.ru (A.B.); drprotsenko@me.com (D.P.); 4Section on Intercellular Interactions, Eunice Kennedy Shriver National Institute of Child Health and Human Development, National Institutes of Health, 29B Lincoln Dr., Bethesda, MD 20892, USA; fitzgerald@cc1.nichd.nih.gov (W.F.); margolil@mail.nih.gov (L.M.); 5Clinical City Hospital №52, Moscow Department of Healthcare, 3 Pekhotnaya Str., 123182 Moscow, Russia; daria_fomina@mail.ru (D.F.); lysenkoma@zdrav.mos.ru (M.L.); gkb52@zdrav.mos.ru (E.M.); 6Department of Cardiology, A.I. Yevdokimov Moscow State University of Medicine and Dentistry, 20 Delegatskaya Str., 127473 Moscow, Russia

**Keywords:** COVID-19, inflammation, cytokine, interleukin, cytokine storm syndrome, tocilizumab

## Abstract

Coronavirus disease 2019 (COVID-19) is characterized by immune activation in response to viral spread, in severe cases leading to the development of cytokine storm syndrome (CSS) and increased mortality. Despite its importance in prognosis, the pathophysiological mechanisms of CSS in COVID-19 remain to be defined. Towards this goal, we analyzed cytokine profiles and their interrelation in regard to anti-cytokine treatment with tocilizumab in 98 hospitalized patients with COVID-19. We performed a multiplex measurement of 41 circulating cytokines in the plasma of patients on admission and 3–5 days after, during the follow-up. Then we analyzed the patient groups separated in two ways: according to the clusterization of their blood cytokines and based on the administration of tocilizumab therapy. Patients with and without CSS formed distinct clusters according to their cytokine concentration changes. However, the tocilizumab therapy, administered based on the standard clinical and laboratory criteria, did not fully correspond to those clusters of CSS. Furthermore, among all cytokines, IL-6, IL-1RA, IL-10, and G-CSF demonstrated the most prominent differences between patients with and without clinical endpoints, while only IL-1RA was prognostically significant in both groups of patients with and without tocilizumab therapy, decreasing in the former and increasing in the latter during the follow-up period. Thus, CSS in COVID-19, characterized by a correlated release of multiple cytokines, does not fully correspond to the standard parameters of disease severity. Analysis of the cytokine signature, including the IL-1RA level in addition to standard clinical and laboratory parameters may be useful to define the onset of a cytokine storm in COVID-19 as well as the indications for anti-cytokine therapy.

## 1. Introduction

Severe acute respiratory syndrome coronavirus 2 (SARS-CoV-2) caused a pandemic of a coronavirus disease in 2019 (COVID-19) and an urgent need for the development of new treatment approaches [1]. One of the main pathophysiologic mechanisms defining the disease’s severity is the overactivation of the immune system associated with an excessive release of cytokines and chemokines, also referred to as “cytokine storm syndrome” (CSS) [2,3]. This type of pronounced immune system reaction is not unique for SARS-CoV-2 infection but has also been described in other clinical situations, such as cytokine release syndrome (CRS) in patients receiving chimeric antigen receptor (CAR)-T cell or bispecific T cell-engaging antibody therapy [4,5], familiar hemophagocytic lymphohistiocytosis (HLH) [6,7], secondary HLH due to malignancies, and autoimmune disorders (macrophage activation syndrome, MAS) [8,9,10], as well as systemic inflammatory response syndrome (SIRS) and acute respiratory distress syndrome (ARDS) in other infectious diseases [11,12,13,14]. In the case of COVID-19, CSS can develop within a few days after the disease’s onset, resulting in a significantly higher rate of patients’ in-hospital complications and mortality [15,16,17].

Despite the negative impact of CSS on the prognosis in patients with COVID-19, the diagnostic criteria of its onset remain unclear. The standard clinical criteria of CSS for noninfectious diseases, such as CRS or HLH, cannot be fully applied to COVID-19 since they also include such parameters as fever and hypoxemia, induced by the viral dissemination even before the beginning of a cytokine storm [5,18,19]. Moreover, many laboratory parameters change in the course of COVID-19, in particular, during the initial phase of the viral infection, thus making them unreliable for the prediction of CSS onset [20,21].

Nevertheless, similar to patients with CRS after CAR-T therapy, primary HLH, or non-COVID-19-related ARDS and SIRS [22,23,24,25,26], COVID-19 patients with CSS are often treated with anti-cytokine therapies. However, in COVID-19-related cytokine storm, the levels of cytokines were found to be significantly lower than those observed in the other above-mentioned pathologies. Therefore, the specific effects of such treatment in COVID-19 patients remain to be elucidated. 

Although clinical trials have revealed that immunomodulatory therapy can be a promising treatment option in COVID-19-associated CSS, the results of these trials are highly controversial [27,28,29,30,31,32,33,34,35]. Discrepancies in trial outcomes can be associated with the different criteria used as indicators for tocilizumab therapy. Indeed, in some trials, HLH parameters, such as fever, leukopenia, increased ferritin, and LDH levels, were used, while others based the therapy solely on increased inflammatory markers, such as high-sensitivity C-reactive protein (hs-CRP) [21,36,37,38]. However, while these parameters correspond well to the development of multiorgan dysfunction and worse clinical prognosis, they do not always correlate with the patterns of hypercytokinemia that can be treated with targeted immunosuppression. Hence, identifying the key cytokine markers of CSS induced by SARS-CoV-2 infection is of utmost importance to provide more reliable indicators for specific immunomodulatory treatments. The identification of these parameters was the goal of the current study.

Specifically, we evaluated the cytokine/chemokine profile in hospitalized COVID-19 patients. We identified the cytokine clusters in mildly and severely ill patients and revealed critical laboratory parameters linked to CSS in severe COVID-19. Overall, our results provide an insight into the cytokine cascade in COVID-19 patients and suggest that IL-6 receptor blockage may be exploited as a rational strategy to suppress SARS-CoV-2–induced CSS when used according to the adapted laboratory criteria of CSS.

## 2. Results

### 2.1. Cytokine Clusterization at Admission

The analysis of laboratory parameters and cytokine concentrations was performed within the first 5 days after the admission of patients (in patients receiving immunomodulatory therapy, within 24 h before the first administration of the treatment), timepoint 1, and the second time within 3–5 days after the first blood test, timepoint 2. While performing the initial clustering analysis of cytokines in all patients, we found that patients at timepoint 1 were separated into two major clusters according to their cytokine profile (including two patients performing as outliers and, therefore, removed from further analysis): cluster 1 was characterized by the mild production of cytokines, compared with cluster 2 with a systemic elevation of most of the cytokines. The concentrations of almost all the cytokines (measured as normalized fluorescence levels) were significantly higher in cluster 2, except for MDC, the concentrations of which did not differ significantly between the two clusters (Figure 1, Appendix A). This clusterization remained unchanged after the exclusion of patients with concomitant use of steroids (Appendix A).

A comparison of the clusters with clinical characteristics by Fisher test revealed that patients assigned to cluster 2 at timepoint 1 more often received any kind of oxygen supplementation therapy and had a higher grade of COVID-19 pneumonia as shown on a CT scan. Moreover, we found a statistically significant correlation between patients’ assignment to cluster 2, the use of tocilizumab, and disease severity (Table 1). Patients in cluster 2 were additionally characterized by changes in the main laboratory markers of CSS that have previously been shown for COVID-19 patients in other studies [39,40,41,42,43,44,45]: these are decreased total protein level, increased white blood cell and neutrophil counts, and increased AST, LDH, ferritin, triglycerides, fibrinogen, PT, and hs-CRP levels (Table 2). 

Thus, according to the general cytokine profile, a subgroup of patients with cases of overt CSS could be distinguished from the patients without CSS by the major changes in several laboratory markers and the severity of the clinical course of the disease. Moreover, the cytokine-based clusters of patients also significantly differed due to the utilization of the tocilizumab therapy, which was administered without randomization to the more severe patients, who were mostly assigned to cluster 2. 

### 2.2. Comparison of Patients with and without Tocilizumab Therapy

Next, we stratified the patients not by their cytokines but by the tocilizumab therapy, applied without randomization, and compared the standard clinical and laboratory parameters. We found that a subgroup of patients where tocilizumab was used had more pronounced changes in parameters of clinical severity but less significant changes in laboratory markers compared to a patient subgroup stratified by the initial upregulation of cytokines (Table 3 and Table 4; see also Table 1 and Table 2 for comparison). In further analysis, we performed a separate assessment of the laboratory parameters and clinical outcomes in these two subgroups of patients.

### 2.3. Correlation of Cytokine Levels at Admission with the Combined Endpoint

At the next step, we compared the concentrations of all cytokines (as normalized fluorescence levels) at timepoint 1 in groups of patients with and without the development of clinical endpoints. We performed this analysis for all patients and separately for the subgroup of patients treated with tocilizumab (Table 5). Using the Mann–Whitney test, we detected significantly higher concentrations of 18 cytokines in the total group of patients who developed clinical endpoints compared to patients without endpoints, corresponding to their disease severity and development of CSS. Moreover, concentrations of IL-6, IL-1RA, IL-10, and G-CSF demonstrated the most prominent differences between these patient subgroups. In patients without tocilizumab therapy, most of these cytokines also retained prognostic significance. At the same time, concentrations of only two of the listed cytokines (IL-10, IL-1RA) correlated significantly with the development of the combined endpoint in the subgroup of patients with tocilizumab therapy, while for IL-6, the correlation with clinical endpoints in this subgroup did not reach statistical significance. 

For the first four cytokines with the most prominent differences between subgroups with and without clinical endpoints, we next explored their absolute concentration levels. We found that statistically significant differences between patients with and without clinical endpoints for absolute cytokine concentrations remained similar to results found for their normalized fluorescence values (Figure 2 and Figure 3).

To establish clinically significant thresholds associated with the combined endpoint, we performed ROC analysis for the aforementioned four cytokines. Assessment of the subgroup consisting of only patients with tocilizumab therapy and of the group of all patients provided similar results for IL-10 and IL-1RA (Figure 3, Appendix A for patients with tocilizumab therapy). Thus, for all patients, we defined threshold levels for the concentration of IL-6 as 39 pg/mL, IL-10 as 57 pg/mL, IL-1RA as 156 pg/mL, and G-CSF as 351 pg/mL. Moreover, we found that the highest specificity and sensitivity for the development of clinical endpoints was achieved when the concentration of IL-6 was used as a threshold (Figure 3). Patients with an initial concentration of IL-6 higher than 39 pg/mL have the highest probability of clinical endpoint developments that correspond well to the earlier data [46,47,48,49,50].

At the same time, IL-6 levels in the subgroup of patients receiving tocilizumab therapy without randomization were initially significantly higher than in the all-patient group and could not be used for stratification of the clinical endpoints (Table 5, Appendix A). We assume that the IL-6 concentration in this subgroup did not correlate with the prognosis because of the leveling up of the IL-6 negative impact caused by the blockage of its receptor with tocilizumab.

### 2.4. Correlation of Laboratory and Clinical Parameters at Admission with the Combined Endpoint

Next, we compared the quantitative laboratory and clinical parameters of CSS separately at timepoint 1 in the groups of patients with and without clinical endpoints. We performed this analysis among all patients (Table 6) and separately among patients treated with tocilizumab (Table 7).

We found that several laboratory parameters at admission correlated with the development of the combined endpoint, including neutrophil count, total protein, ALT and AST, hs-CRP, oxygen saturation, and respiratory rate. These results were expected since most of these parameters were found in earlier studies to define the disease severity and to correlate with mortality in COVID-19 [40,41,45,51,52]. However, while having a significant effect on the combined endpoint in the all-patient group, the level of hs-CRP did not correlate with the prognosis in the subgroup of patients with tocilizumab therapy. At the same time, tocilizumab therapy itself did not decrease the risk of combined endpoint development; probably because of the worse initial characteristics of these patients compared to patients without tocilizumab therapy. 

### 2.5. Correlation of Cytokine Levels at Admission with Laboratory and Clinical Parameters of CSS

When we calculated Spearman correlations between the cytokine concentrations and standard laboratory and clinical parameters at timepoint 1, we found, surprisingly, no strong correlations of cytokine concentrations with any evaluated laboratory or clinical parameters, except the hs-CRP level (Appendix A). Majorly strong statistically significant correlations (*p*-value < 0.05, coefficient of correlation >0.5/<−0.5) were found between the hs-CRP level and the concentrations of two of the four cytokines with the most prominent prognostic impact, IL-6 and G-CSF (Figure 4), as well as between hs-CRP level, Il-1β, TNFα, and MIP-1α (Appendix A). We also found significant correlations between hs-CRP level and the levels of fibrinogen and ferritin (Appendix A).

We chose hs-CRP level, which was the only parameter correlating with the levels of two of the four cytokines with a prognostic impact and, using ROC analysis, established clinically significant thresholds for its level that were associated with the development of clinical endpoints (Figure 5). Thus, we identified a threshold level of 108 mg/L with a 67% specificity and 64% sensitivity that correlated with the development of clinical endpoints. This threshold level is in agreement with the threshold of 100 mg/L used in the previous meta-analyses [53], but not with the threshold of 75 mg/L used as an indication for tocilizumab therapy in the recent clinical trials [31]. However, the hs-CRP threshold of 75 mg/L is very close to the levels received in our subgroup of patients without tocilizumab therapy, thus making it an appropriate criterion to be used for the identification of the CSS (Appendix A). 

### 2.6. Dynamic Changes in Cytokine Concentrations after the Treatment with Tocilizumab

In the last part, we evaluated the changes in cytokine concentration between timepoint 1 and timepoint 2 and investigated whether treatment with tocilizumab affected these changes (Table 8). Most cytokines demonstrated either no change or a statistically significant decrease in concentration for all groups (total patient, patients without tocilizumab therapy, patients with tocilizumab therapy), with three exceptions. The only cytokine that statistically significantly increased during hospitalization in all three patient groups was MIP-1β, whereas Eotaxin also increased in all three groups but this change reached statistical significance only in the group of patients treated with tocilizumab. Moreover, the concentration of IL-6 decreased in the group of patients without tocilizumab therapy while increasing in the group of patients treated with tocilizumab, in agreement with the consequence of the IL-6 receptor blockage by the drug. The other major significant changes (>0.2 median log2 fold change in normalized fluorescence) were recorded for the concentrations of IL-10 and IP-10, which decreased in all groups of patients during the follow-up; for the concentrations of G-CSF, which decreased in the all-patient group and separately in patients without tocilizumab therapy; and for the concentrations of IL-1RA, which decreased in the all-patient group and in patients treated with tocilizumab. 

Finally, we compared the changes in the four aforementioned cytokines with the most prominent prognostic impact during the follow-up in groups with and without a combined endpoint (Figure 6). As with the all-patient groups, we found that IL-6 concentration increased in the groups of patients with tocilizumab therapy accordingly with its effect as compared to patients without tocilizumab. The concentration of IL-10 decreased in all-patient groups independently from the tocilizumab therapy. The concentration of G-CSF, while decreasing in patient groups without clinical endpoints, showed almost no changes in patients with clinical endpoints, with no dependence on the tocilizumab therapy. In contrast, the concentration of IL-1RA behaved differently in groups with clinical endpoints with and without tocilizumab therapy: it increased in patients with clinical endpoints without tocilizumab and decreased in patients with clinical endpoints with tocilizumab therapy. These changes might add an explanation for the neutral results of the studies on direct IL-1R antagonists in COVID-patients, where the initial cytokine concentration was not taken into consideration [54]. 

In conclusion, we showed that the concentration of IL-1RA remained the only prognostically significant factor for both groups of patients with and without tocilizumab therapy. Potentially, it can be used as an additional indication criterion for the administration of anti-cytokine therapy. 

## 3. Discussion

Approximately 80% of severe and moderate COVID-19 cases are accompanied by the development of CSS due to excessive immune activation [55,56,57,58,59]. It is associated with a systemic release of cytokines [60,61], causing multiorgan failure resulting in decreased patient survival [62,63,64]. This syndrome has already been described for patients receiving CAR-T therapy and other anticancer treatments [4,5,65] as well as in HLH syndromes [66,67], and different bacterial and viral infections [13,68,69] including other coronaviruses [70]. However, the laboratory signature of CSS in COVID-19 differs significantly from those of the other pathologies [59]. In particular, the levels of pro-inflammatory cytokines were shown to be significantly lower in COVID-19 patients than in critically ill patients with ARDS or sepsis [23,26]. Moreover, the profile of cytokine changes depends on the specific disease, with IFN-γ playing the key role in primary HLH [71], IL-18 in MAS [72], and IL-6 in CRS [73]. Even though it is caused by viral infections, the cytokine storm is characterized by different cytokine signatures in the cases of MERS [74,75,76], SARS [77,78], and influenza [20,79,80]. Thus, although it is regarded as a similar pathological state, the treatment of different CSSs has to be individualized according to the degree and difference in the cytokine cascade activation.

Despite the importance of the treatment strategy, there are still no unified criteria to diagnose cytokine storm syndromes. While the standard parameters defining CRS after specific T cell antibody therapy mostly include clinical characteristics of the patients, the criteria for HLH include laboratory markers such as cytopenia, elevated liver enzymes, ferritin, and CRP, as well as changes in coagulation factors [4,81]. It was shown that not all of these markers can be used as diagnostic criteria for the development of COVID-19-associated CSS. According to the initial trials, patients with severe and fatal disease showed an increased leukocyte count, decreased lymphocyte and platelet counts, increased biomarkers of inflammation, cardiac and muscle injury, liver and kidney function, and coagulation [39,40,41,43,45,52,82,83,84,85]. However, in the further larger studies, only a few of these markers were found to be independently associated with increased mortality [51]. 

In analyses of the cytokine storm severity in COVID-19, most of the trials used the CRP level as a unified marker of inflammation. It was shown to be a significant prognostic marker for disease severity as well as mortality in COVID-19. Specifically, an hs-CRP level threshold of 91 mg/L was reported to be associated with the severe form of the disease [86], while, in other studies, lower values of 66 mg/L or even 33 mg/L were found to be significant for patients’ prognosis [87,88]. In our trial, we found that a CRP level higher than 108 mg/L was linked to the mortality of the patients, which is in agreement with the threshold of 100 mg/L found in the published meta-analyses [53]. However, when we assessed the interaction between the multi-cytokine signature of the storm and hs-CRP level, we found only a minor number of significant correlations. Moreover, clinical and laboratory criteria showed no strong correlations with cytokine levels. These data indicate the need for a further search of a better criterion or combination of criteria to define the cytokine storm development in COVID-19.

In research aimed at establishing the cytokine signature of COVID-19, several trials investigated clusters of cytokines to assess the development of the cytokine storm. In these trials, many CSS-related cytokines showed an increase corresponding to the severity of the disease [89], providing proof of a severe general inflammatory activation in COVID-19 [17,90,91]. Moreover, these cytokine levels correlated with each other, showing the concordance of the inflammatory changes during the CSS [92]. The previously established cytokine clusters included growth factors, mediators of tissue repair, immune effector modulators, and chemoattractants. 

In our study, we assessed the development of the cytokine storm according to the clusters of circulating cytokines in the blood of the patients with proven COVID-19 within the first days after admission to hospital, before the initiation of the antiinflammatory therapy. While performing the randomization, we found that the patients could be separated into two distinct clusters according to their cytokine levels. Moreover, these clusters corresponded to the severity of the disease. We also showed that the clusterization of the cytokines according to the cytokine storm severity corresponded to the changes in many of the standard parameters, including leukocyte and neutrophil counts, levels of total protein, AST, LDH, ferritin, triglycerides, fibrinogen, prothrombin time, and hs-CRP, that were shown in previous trials to be associated with the disease severity or mortality. However, while it was applied without initial randomization, tocilizumab was used in the patients with more severe changes in clinical parameters but less pronounced laboratory changes, thus indicating an underestimation of the cytokine storm development in some patients with a less severe clinical state and the need for more specified laboratory markers to initiate the specific anti-cytokine therapy.

Many of the individual cytokines from the clusters correlated with the disease severity and mortality according to the recent trials, including TNF-α, IL-10, IL-15, IL-12, IL-2, IL-6, IFN-α, IFN-γ, and IL-1RA, as well as many chemokines [91,92,93,94,95]. Later studies underlined the significance of the other cytokines, such as IL-1β, sIL-2Rα, IL-17, IL-18, MCP-3, M-CSF, MIP-1α, G-CSF, IP-10, and MCP-1 [17,39,96,97]. IL-1β, IL-1RA, IL-6, IL-7, IL-10, IP-10, and TNF-α showed the strongest correlation with disease severity in several recent trials [97]. Despite being lower than in the other cases of cytokine storm, i.e., HLH or CRS, these cytokines still indicate a worse prognosis for COVID-19 patients [23]. 

Unfortunately, the data concerning the clinical impact of different cytokines remain highly controversial [49,50,98,99,100]. For example, the levels of IL-6 were shown in several studies to be the main contributor to the prognosis in COVID-19 [101,102,103,104,105,106,107], but at the same time, the threshold of IL-6 levels differed in these trials significantly, with a range from 30 pg/mL to 163.4 pg/mL [108]. In our study, we found that a threshold of 39 pg/mL corresponded to the disease severity and mortality of the patients. However, we found no significant correlation of the IL-6 levels with clinical endpoints in the subgroup of patients treated with tocilizumab. We assume that the effect of IL-6 was masked because of the initial higher levels in this group of patients (with a threshold of 62 pg/mL) as well as due to the impact of the tocilizumab treatment, targeted against IL-6 receptors. Therefore, despite the general findings indicating the usefulness in measuring the serum IL-6 level [46], its clinical implication to guide anti-cytokine treatment is still lacking a prospectively established threshold. 

Many hopes were invested in the application of direct anti-cytokine therapy in COVID-19 patients, especially of the IL-6 receptor blocker tocilizumab, which was previously widely used in CRS caused by CAR-T therapy [109,110]. Several trials reported a positive effect of tocilizumab therapy on the symptoms and respiratory function, as well as on mortality in COVID-19 patients [111]. Despite showing heterogeneous results in the following randomized clinical trials [27], IL-6 inhibitors were included in the current NIH COVID-19 guidelines for the treatment of severe and critical COVID-19.

In the recent meta-analysis, the most pronounced benefit of tocilizumab was shown in patients with hs-CRP levels greater than 100 mg/L, while it had no positive effect in patients with hs-CRP levels lower than 100 mg/L [53]. In our study, we found no significant impact of hs-CRP levels on the prognosis in patients treated with tocilizumab, while there was a threshold hs-CRP level of 158 mg/L in that subgroup. On the other hand, in patients not receiving tocilizumab, the hs-CRP level threshold of 78 mg/L was statistically significant in defining the prognosis. For the all-patient group this threshold constituted 108 mg/L, corresponding to the commonly used threshold for the application of the anti-cytokine therapy in current clinical trials [31]. Thus, we assume that an extreme elevation of the hs-CRP level higher than 158 mg/L might reflect the beginning of multiorgan damage, to which the targeted therapy with tocilizumab brings no additional benefit, while approximate values between 78–158 mg/L can be used as an indication for the initiation of the immunomodulatory therapy. 

In our study, not only did we investigate the effect of tocilizumab on the prognosis of patients, but we also tried to identify the main cytokines that can predict the effectiveness of this therapy. We found that while many cytokines, including IL-6, showed an impact on the prognosis of COVID-19 patients, only two of them, IL-1RA and IL-10, were of clinical significance in the subgroup of patients receiving tocilizumab therapy. At the same time, IL-6 had a borderline significance for the prognosis in that subgroup of patients. The importance of IL-6 and IL-10 as major pro- and antiinflammatory cytokines was shown in many trials on COVID-19 [49,50,98,99,112,113] with meta-analyses showing the prevalent role of these two cytokines in disease severity and prognosis [106]. However, when analyzed according to the time from disease onset, the IL-6 elevation was found only at the late stage of severe COVID-19 while IL-10 and IL-1RA levels were significantly associated with disease severity and patients’ outcomes already at the first week after symptom onset [114]. 

Moreover, according to our data, IL-1RA was the only cytokine that showed a prognostic significance after separation into groups of patients with and without tocilizumab therapy as well as a significant dynamic decrease in patients receiving tocilizumab, especially in those with the worse prognosis. IL-1RA was shown to control inflammatory responses during the early stages of immune activation [115] while binding to the IL-1R and modulating the production of IL-1 and type I IFN [116], two important cytokines involved in the early phase of coronavirus infection [117,118]. Finally, because of the difference in the IL-1RA serum concentrations in severe and mild COVID-19, it is assumed that higher levels of IL-1RA observed in severe cases suggest an overactive immune response, which may contribute to the inflammation-induced tissue damage and therefore correlate well with the prognosis [114]. 

Overall, most of the individual cytokines in our study showed no significant effect on the prognosis of patients receiving tocilizumab therapy. Therefore, we assume that a cytokine signature including several potent individual cytokines (i.e., Il-6, IL-1RA, and IL-10) in addition to the standard laboratory markers might be a more useful tool in the stratification of patients with COVID-19 and CCS that will benefit from the immunomodulatory therapy. 

## 4. Materials and Methods

We performed a non-randomized observational trial of patients admitted to Davydovsky Moscow City Clinical Hospital and to Moscow Clinical Hospital №40 from April to December 2020 with an initial diagnosis of COVID-19 infection. During this time period, we included prospectively 140 patients, whom we followed up during the hospitalization and analysed their outcomes in correlation to their clinical and laboratory characteristics as well as therapy received. 

Inclusion criteria were: positive results of polymerase chain reaction (PCR) or IgM measurement for SARS-CoV-2;COVID-19 pneumonia defined by computed tomography (CT);moderate or severe course of the disease (criteria are summarized in Table 9 [119]);signed informed written consent;age > 18.

Exclusion criteria were:presence of acute or chronic infectious diseases other than COVID-19;presence of acute and chronic systemic inflammatory diseases or any type of cancer;negative PCR or antibody test for SARS-CoV-2.

Upon hospitalization, patients were treated according to the Russian National Clinical COVID-19 Recommendations, including IL-6 receptor blocking antibody tocilizumab and steroids. Of the 140 patients included because of the presence of COVID-19 symptoms, 11.4% further tested negative for SARS-CoV-2 by PCR or antibody test and therefore were excluded from the analysis. During the follow-up period, we further excluded all patients receiving any other immunomodulatory therapy, such as other anti-cytokine drugs or antibiotics with immunomodulatory effect, cytokine adsorption techniques, or reconvalescent plasma. After exclusion of patients with non-specified immunomodulatory therapy, we performed an analysis of 98 patients receiving either no immunomodulatory therapy (*n* = 46) or tocilizumab therapy (*n* = 52) together and in separated subgroups with and without anti-cytokine therapy. We also performed a separate analysis of patients dependent on the utilization of steroid therapy. General patient characteristics are presented in Table 10.

In all patients, we assessed standard clinical characteristics, such as grade of pneumonia on CT scan, temperature, respiratory rate, and oxygen saturation, as well as standard laboratory parameters, including complete blood count (hemoglobin level, erythrocyte count, platelet count, white blood cell, and lymphocyte/neutrophil count), biochemical parameters (total protein, bilirubin, creatinine, glucose, aspartate aminotransferase (AST), alanine aminotransferase (ALT), lactate dehydrogenase (LDH), ferritin, triglycerides), coagulation parameters (fibrinogen, D-dimer, prothrombin time (PT)), and hs-CRP. Blood samples collected in EDTA tubes during the standard blood testing were further used for a multiplexed assessment of 41 cytokines with a bead-based assay. Blood tests were assessed twice: the first time within the first 5 days after admission (in patients receiving immunomodulatory therapy, within 24 h before the first administration of the treatment), timepoint 1, and the second time within 3–5 days after the first blood test, timepoint 2. 

### 4.1. Blood Collection

The utilization of blood samples for research purposes was approved by the Moscow city ethics committee. To obtain plasma for the cytokine analysis we used blood samples collected in vacuum tubes with EDTA (Sarstedt, Nuembrecht, Germany). After collection, blood samples were centrifuged in a bucket rotor centrifuge for 10 min at 1000× *g* at room temperature (RT). We transferred the upper plasma layer into a new tube while leaving approximately 1 mL of plasma above the blood cells intact to reduce cellular contamination. Transferred plasma was pipetted thoroughly, aliquoted by 300 µL, and stored at −80 °C until further analysis.

### 4.2. Cytokine Measurement

Forty-one cytokines in blood were measured with a commercial kit MILLIPLEX MAP Human Cytokine/Chemokine Magnetic Bead Panel (Merck Millipore, Burlington, MA, USA). The cytokine panel included interleukin-1α (IL-1α), IL-1β, IL-1RA, IL-2, IL-3, IL-4, IL-5, IL-6, IL-7, IL-8, IL-9, IL-10, IL-12 (p40), IL-12 (p70), IL-13, IL-15, IL-17A, fractalkine (CX3CL1), growth-regulated alpha (GRO-α or CXCL1), interferon-γ-induced protein-10 (IP-10 or CXCL10), monocyte chemoattractant protein-1 (MCP-1 or CCL2), MCP-3 (CCL7), macrophage inflammatory protein-1α (MIP-1α or CCL3), MIP-1β (CCL4), regulated on activation normally T cell expressed and secreted (RANTES or CCL5), eotaxin (CCL11), macrophage-derived chemokine (MDC or CCL22), soluble CD40-ligand (sCD40L), epidermal growth factor (EGF), fibroblast growth factor-2 (FGF-2), Fms-like tyrosine kinase 3 ligand (Flt-3L), vascular endothelial growth factor (VEGF), granulocyte colony-stimulating factor (G-CSF), granulocyte-macrophage colony-stimulating factor (GM-CSF), platelet-derived growth factor-AA (PDGF-AA), PDGF-AB/BB, transforming growth factor-α (TGF-α), interferon-α2 (IFN-α2), IFN-γ, and tumor necrosis factor-α (TNF-α) and TNF-β. 

The standard curve was built up from 8 standard dilutions in triplicate, with the 1st–3rd standard dilutions with dilution factor 5 and the 4th–8th dilutions with dilution factor 4. We used a serum matrix diluted in an assay buffer to mimic the matrix effect on the standard curve, controls, and blank wells. Thus, 25 µL of standards and controls were diluted with 25 µL of serum matrix. Plasma was diluted 4 times in an assay buffer to reduce the matrix effect and added in a volume of 50 µL to each well. We added 15 µL of 41-plex magnetic beads to each well and incubated for 18 h at 4 °C. Beads were washed twice with an automatic magnetic washer (Biotek ELx405, Winooski, VT, USA) and incubated with detection antibodies for 1 h at 25 °C. Antibodies were diluted with wash buffer 1.93 times and added in the amount of 25 µL per well. After incubation, we added 15 µL of Streptavidin-PE solution to each well and incubated the final solution for 30 min at 25 °C. Then, beads were washed twice, resuspended in the sheath fluid, and analyzed with a Luminex 200 instrument (Luminex Corp, Austin, TX, USA). For the analysis, we collected 100 beads per region. During the analysis, we used 5PL fit for the standard curve. We excluded RANTES from the analysis because of a high inhibition level.

### 4.3. Clinical Endpoints Assessment

We analyzed a short-term combined clinical endpoint for all the patients as:application of high-flow oxygen therapy or noninvasive/invasive lung ventilation;patient transfer to intensive care;in-hospital mortality.

### 4.4. Statistical Analysis

We performed statistical analysis with Python-3. The expression values obtained in the present study were in most cases not normally distributed according to the Shapiro–Wilk test, and therefore are represented as medians and interquartile ranges (Q25–Q75). Age is presented as mean and standard deviation. Since distributions were not normal, for comparison of several groups we used the Mann–Whitney rank test with continuity correction. For the analysis of categorical parameters, we used a two-tailed Fisher’s Exact Test with 2 × 2 frequency tables. To overcome errors from multiple comparisons we performed a Benjamini–Hochberg FDR correction with a calculation of critical values for each comparison matched with corresponding *p*-values; we calculated adjusted *p*-values and compared them with a critical value of 0.05; below, “*p*-value” refers to Benjamini–Hochberg-adjusted *p*-values, if not stated otherwise. For a heatmap data visualization, *p*-values are shown as log10 from original *p*-values, where log10 *p*-values ≤ 1.3 correspond to original *p*-values < 0.05. For the age distribution, we made the assumption of its normality and analyzed this distribution using the *t*-test. 

In many cases, cytokine levels were outside the limits of detection of the Luminex device, which did not allow their quantitative analysis: 12.5% of all measurements were either below or above limits of detection, with more than 25% of values systematically missing for several cytokines. Therefore, following the published method [120], for assessing the cytokine levels we additionally used log2 of pre-normalized fluorescence intensity values. These values were used for quantitative comparison of cytokine levels between different subgroups of patients; it was additionally *z*-score normalized and used as a basis for hierarchical clustering analysis. For the assessment of clinically significant thresholds associated with clinical endpoints, we performed ROC analysis; however, because of the size of the experimental group, we did not split the data into training and testing subsets.

## 5. Conclusions

In conclusion, our data indicate the importance of the evaluation of a cytokine signature, including, especially, IL-1RA levels, in addition to the CPR level and other standard clinical and laboratory parameters to develop a better prediction strategy for the development of a cytokine storm in COVID-19 as well as to assess the indications for anti-cytokine therapy. 

## Figures and Tables

**Figure 1 ijms-23-07937-f001:**
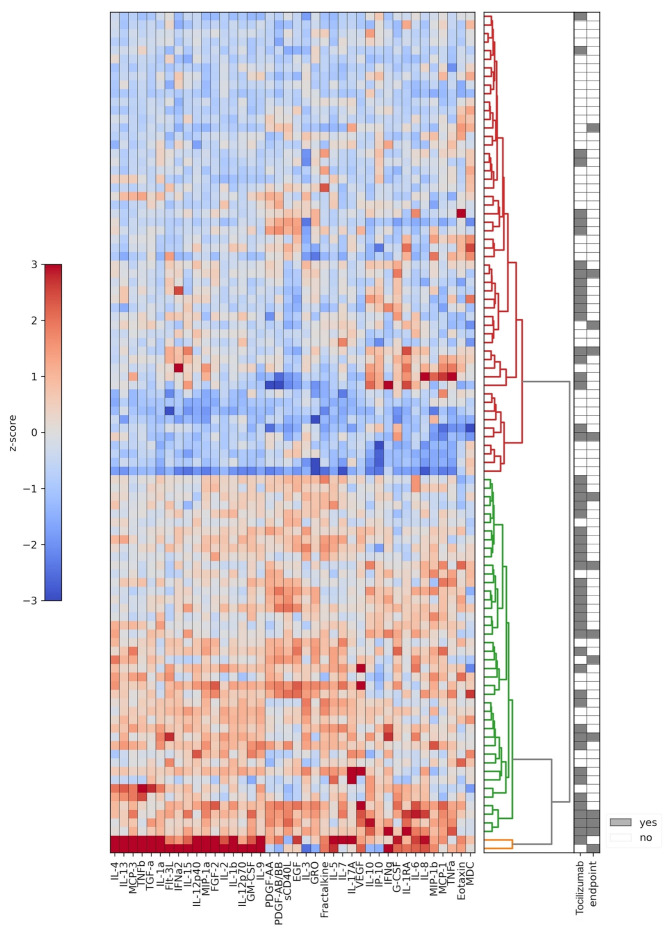
All patients’ cytokine clusterization at timepoint 1. Data are presented as log 2 of the normalized fluorescence intensity values that were further z-score normalized and used as a basis for hierarchical clustering analysis. Cluster 1 is indicated with the red hierarchical tree, cluster 2 is indicated with the green hierarchical tree, orange color indicates two outliers that were excluded from the further analysis. Tocilizumab indicates application of the immunomodulatory therapy. Endpoint indicates a presence of a combined clinical endpoint, which included application of high-flow oxygen therapy or noninvasive/invasive lung ventilation, patient transfer to intensive care, and in-hospital mortality.

**Figure 2 ijms-23-07937-f002:**
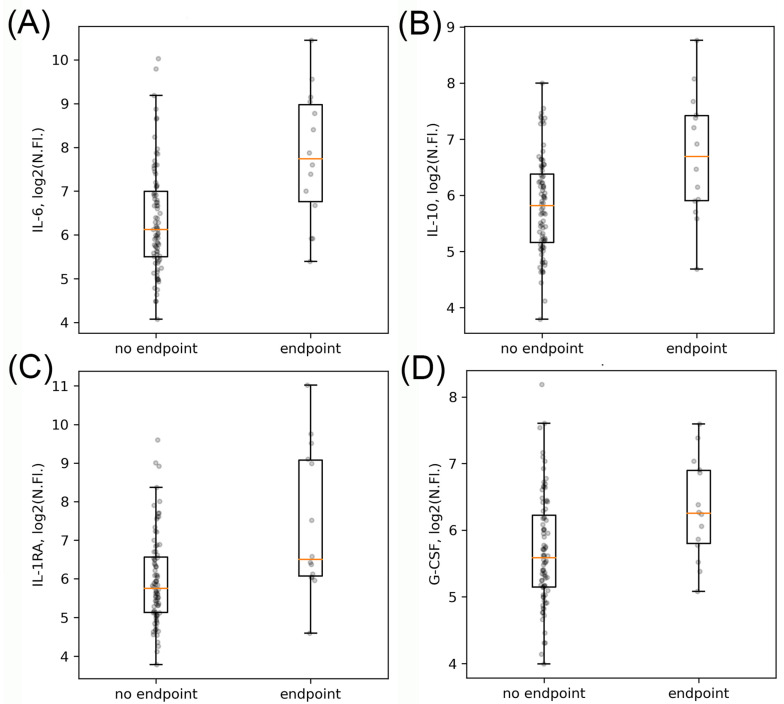
Boxplots for the concentrations of four cytokines prominently associated with clinical endpoints in all patients. IL-6 (**A**). IL-10 (**B**). IL-1RA (**C**). G-CSF (**D**). Data are presented as log2 of the normalized fluorescence values.

**Figure 3 ijms-23-07937-f003:**
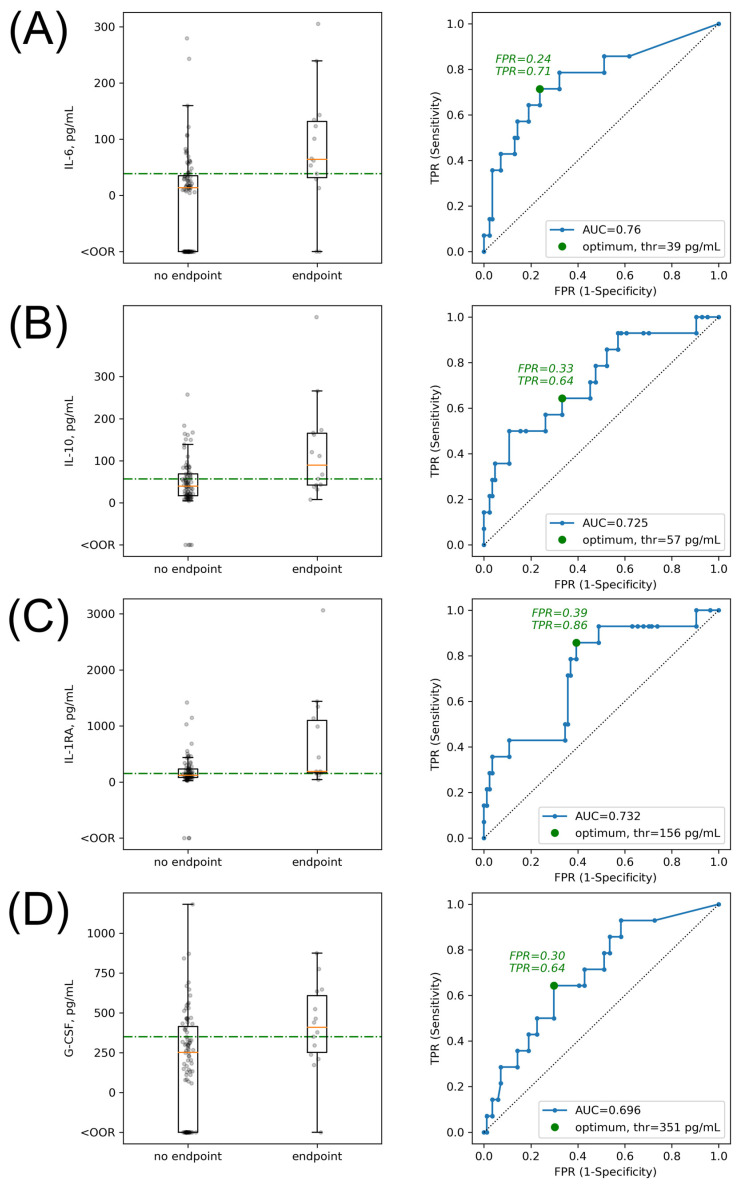
Boxplots for the absolute concentrations and ROC curves for selecting optimal concentration threshold for the 4 cytokines prominently associated with clinical endpoints. IL-6 (**A**). IL-10 (**B**). IL-1RA (**C**). G-CSF (**D**). OOR—out of range (<lower limit of detection).

**Figure 4 ijms-23-07937-f004:**
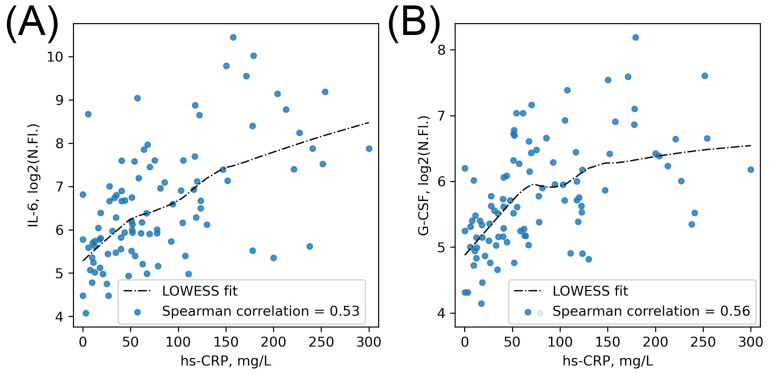
Correlation of cytokine levels at timepoint 1 with CRP level. Il-6 vs hs-CRP (**A**). G-CSF vs hs-CRP (**B**). Data are presented as log2 of the normalized fluorescence values with the Spearman correlations.

**Figure 5 ijms-23-07937-f005:**
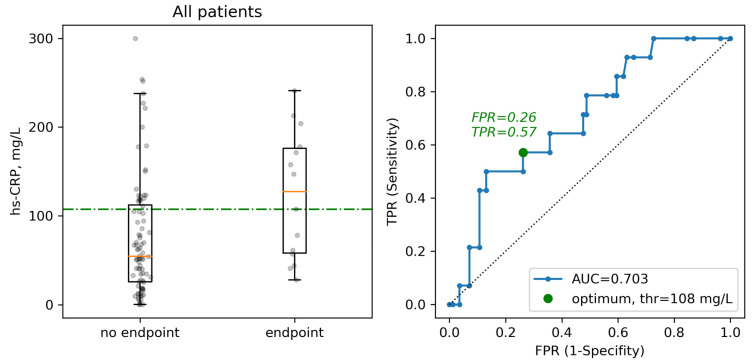
Boxplots and ROC curves for selecting optimal concentration threshold of hs-CRP for defining clinical endpoints.

**Figure 6 ijms-23-07937-f006:**
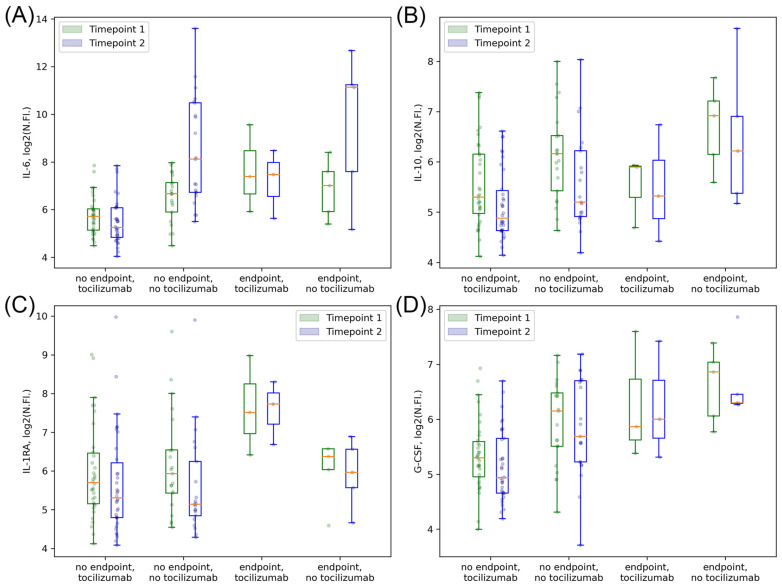
Boxplots for dynamic changes in four cytokines associated with clinical endpoints in patients with tocilizumab. IL–6 (**A**). IL-10 (**B**). IL-1RA (**C**). G-CSF (**D**). Data are presented as log2 of the normalized fluorescence values.

**Table 1 ijms-23-07937-t001:** Comparison of clinical parameters and outcomes between patient cytokine clusters. *p*-values are presented according to the Fisher exact test.

	Cluster 1	Cluster 2	Fisher Test *p*-Value
**Age**	61.0 ± 14.0	58.7 ± 14.2	0.45
**Male sex**	48%	50%	1
**Coronary heart disease**	13%	7%	0.51
**Prior stroke**	11%	5%	0.46
**Prior myocardial infarction**	7%	5%	0.69
**Hypertension**	65%	52%	0.3
**Diabetes mellitus**	17%	31%	0.14
**Chronic obstructive pulmonary disease (COPD) or asthma, %**	4%	10%	0.4
**Chronic kidney disease**	20%	17%	0.79
**Steroid therapy**	33%	24%	0.37
**Tocilizumab therapy**	37%	76%	0.00021
**CT grade > 2**	32%	55%	0.036
**Severe course of the disease**	61%	81%	0.045
**Any oxygen therapy**	33%	60%	0.013
**Transfer to intensive care**	7%	19%	0.12
**High-flow oxygen therapy or noninvasive/invasive lung ventilation**	9%	14%	0.53
**In-hospital mortality**	6%	7%	1
**Combined endpoint**	9%	19%	0.23

**Table 2 ijms-23-07937-t002:** Comparison of laboratory parameters between patients’ cytokine clusters. Data are presented as median and interquartile range [Q25;Q75]. *p*-values are presented according to the Mann–Whitney test.

	Cluster 1 Median	Cluster 1 [Q25;Q75]	Cluster 2 Median	Cluster 2 [Q25;75]	Mann–Whitney *p*-Value
**Respiratory rate**	20	[18.2;22.0]	20	[18.0;22.0]	0.28
**Oxygen saturation**	95	[93.0;96.0]	94	[90.0;95.0]	0.015
**Temperature**	38.5	[38.0;39.0]	38.5	[38.0;39.0]	0.26
**Hemoglobin**	136.5	[121.8;152.0]	130.5	[120.2;141.8]	0.062
**Erythrocyte count**	4.5	[4.1;5.0]	4.5	[4.1;4.8]	0.075
**Platelet count**	197.5	[162.8;252.8]	200	[147.8;250.2]	0.43
**Leucocyte count**	5.6	[4.2;6.9]	6.4	[4.8;8.2]	0.04
**Lymphocyte count**	1.2	[0.8;1.6]	1.2	[0.9; 1.6]	0.89
**Neutrophil count**	3.6	[2.6;5.2]	4.9	[3.2;6.3]	0.0064
**Total protein**	75	[68.0;79.5]	70.4	[67.6;75.0]	0.038
**Total bilirubin**	8.6	[6.5;13.2]	10.3	[7.6;13.2]	0.076
**Creatinine**	103	[90.0;119.8]	100.5	[78.0;123.0]	0.32
**ALT**	41	[26.0;58.0]	42	[35.0;67.8]	0.12
**AST**	34	[25.0;56.8]	58.5	[36.2;72.8]	0.00074
**LDH**	281	[235.5;372.2]	389	[300.2;509.0]	1.30 × 10^−5^
**hs-CRP**	40.7	[18.1;67.8]	105	[57.4;172.9]	2.90 × 10^−6^
**Ferritin**	391	[193.0;648.0]	556	[330.2;1058.2]	0.0066
**Triglycerides**	1.3	[0.9;1.6]	1.6	[1.2;2.1]	0.0041
**Fibrinogen**	5.9	[4.4;6.9]	6.5	[5.3;7.6]	0.027
**Prothrombin time**	11.2	[10.5;12.2]	11.5	[11.0;12.4]	0.072
**D-dimer**	432.5	[237.5;840.5]	461	[297.0;1019.5]	0.16

**Table 3 ijms-23-07937-t003:** Comparison of clinical parameters and outcomes between subgroups of patients with and without tocilizumab therapy. *p*-values are presented according to the Fisher exact test.

	Patients without Tocilizumab	Patients with Tocilizumab	Fisher Test *p*-Value
**Age**	61.5 ± 14.1	59.4 ± 14.7	0.6
**Male sex**	46%	52%	0.55
**Coronary heart disease**	7%	13%	0.33
**Prior stroke**	11%	8%	0.73
**Prior myocardial infarction**	4%	8%	0.68
**Hypertension**	61%	58%	0.84
**Diabetes mellitus**	22%	23%	1
**Chronic obstructive pulmonary disease (COPD) or asthma, %**	7%	6%	1
**Chronic kidney disease**	17%	19%	1
**Steroid therapy**	33%	24%	0.37
**CT-grade > 2**	26%	56%	0.004
**Severe course of the disease**	50%	89%	4.40 × 10^−5^
**Any oxygen therapy**	24%	62%	0.00023
**Transfer to intensive care**	7%.	17%	0.13
**High-flow oxygen therapy or noninvasive/invasive lung ventilation**	7%	15%	0.21
**In-hospital mortality**	7%	8%	1
**Combined endpoint**	9%	19%	0.16

**Table 4 ijms-23-07937-t004:** Comparison of laboratory parameters between subgroups of patients with and without tocilizumab therapy. Data are presented as median and interquartile range [Q25;Q75]. *p*-values are presented according to the Mann–Whitney test.

	Patients without Tocilizumab, Median	Patients without Tocilizumab [Q25;Q75]	Patients with Tocilizumab, Median	Patients with Tocilizumab [Q25;Q75]	Mann–Whitney *p*-Value
**Respiratory rate**	20	[18.0;22.0]	20	[18.0;22.0]	0.31
**Oxygen saturation**	95	[93.0;96.0]	94	[90.0;95.0]	0.014
**Temperature**	38.4	[37.5;38.8]	38.8	[38.0;39.0]	0.014
**Hemoglobin**	136	[122.8;152.2]	132	[117.8;142.0]	0.028
**Erythrocyte count**	4.6	[4.1;5.0]	4.4	[4.0;4.8]	0.053
**Platelet count**	197.5	[160.2;268.0]	203.5	[149.2;251.0]	0.38
**Leucocyte count**	5.7	[4.8;7.4]	6.1	[4.2;7.9]	0.49
**Lymphocyte count**	1.3	[1.0;1.7]	1.0	[0.8;1.4]	0.014
**Neutrophil count**	3.6	[2.9;5.4]	4.3	[2.9;5.9]	0.29
**Total protein**	76	[71.6;80.8]	68.4	[65.0;73.7]	6.50 × 10^−6^
**Total bilirubin**	10.2	[7.5;13.7]	8.9	[6.4;11.3]	0.058
**Creatinine**	101.5	[90.0;122.2]	101.5	[79.5;123.0]	0.36
**ALT**	46.5	[28.0;63.5]	40	[26.8;59.5]	0.15
**AST**	36.5	[25.5;60.0]	46	[32.2;70.2]	0.069
**LDH**	277.5	[242.2;375.5]	364.2	[289.5;470.9]	3.10 × 10^−3^
**hs-CRP**	33.6	[13.4;74.3]	89.2	[53.6;150.8]	4.00 × 10^−6^
**Ferritin**	351	[174.2;618.8]	581	[331.5;850.5]	0.0092
**Triglycerides**	1.3	[1.0;1.6]	1.5	[1.2;2.0]	0.093
**Fibrinogen**	5.8	[4.5;6.8]	6.5	[5.1;7.5]	0.038
**Prothrombin time**	11.4	[10.9;12.3]	11.5	[10.5;12.3]	0.31
**D-dimer**	418.5	[190.5;1006.0]	448	[350.5;857.0]	0.19

**Table 5 ijms-23-07937-t005:** Comparison of cytokine levels between patients with and without combined endpoint in all patients and separately in patients receiving tocilizumab. *p*-values are presented according to the Mann–Whitney test.

	All Patients, *p*-Values	Patients without Tocilizumab, *p*-Values	Patients with Tocilizumab, *p*-Values
**IL-6**	0.00063	0.0053	0.061
**IL-1RA**	0.0015	0.015	0.042
**IL-10**	0.0037	0.25	0.0076
**G-CSF**	0.0051	0.024	0.097
**IL-4**	0.0069	0.017	0.18
**IL-13**	0.0074	0.067	0.15
**TGF-a**	0.013	0.0074	0.25
**VEGF**	0.015	0.045	0.17
**IL-8**	0.016	0.041	0.22
**MCP-3**	0.023	0.032	0.27
**IL-1a**	0.023	0.053	0.16
**MIP-1a**	0.025	0.17	0.13
**IL-7**	0.04	0.24	0.16
**IL-17A**	0.047	0.057	0.22
**GM-CSF**	0.047	0.36	0.14
**IL-15**	0.052	0.2	0.2
**IL-1b**	0.068	0.16	0.28
**IFNg**	0.073	0.45	0.07
**TNFb**	0.078	0.096	0.35
**GRO**	0.084	0.45	0.21
**PDGF-AA**	0.086	0.38	0.16
**FGF-2**	0.093	0.31	0.42
**IL-9**	0.1	0.15	0.48
**PDGF-AB/BB**	0.12	0.3	0.48
**sCD40L**	0.12	0.13	0.31
**IL-12p70**	0.12	0.21	0.41
**MCP-1**	0.14	0.17	0.32
**IP-10**	0.14	0.48	0.22
**MIP-1b**	0.15	0.3	0.32
**Flt-3L**	0.15	0.26	0.29
**Fractalkine**	0.16	0.24	0.5
**MDC**	0.16	0.13	0.43
**IFNa2**	0.18	0.42	0.31
**IL-5**	0.19	0.096	0.24
**IL-2**	0.2	0.42	0.31
**TNFa**	0.27	0.38	0.49
**IL-3**	0.29	0.083	0.22
**IL-12p40**	0.42	0.5	0.34
**Eotaxin**	0.47	0.37	0.48
**EGF**	0.5	0.27	0.28

**Table 6 ijms-23-07937-t006:** Comparison of laboratory and clinical parameters at timepoint 1 between patients with and without combined endpoint in all patients. Data are presented as median and interquartile range [Q25;Q75]. *p*-values are presented according to the Mann–Whitney test.

	All Patients, No Endpoint Median	All Patients, No Endpoint [Q25;Q75]	All Patients, Endpoint Median	All Patients, Endpoint [Q25;Q75]	Mann–Whitney *p*-Value
**Respiratory rate**	20	[18.0;22.0]	22	[20.2;22.0]	0.044
**Oxygen saturation**	94.5	[93.0;96.0]	91.5	[86.0;95.8]	0.04
**Temperature**	38.5	[38.0;39.0]	38.6	[38.0;39.5]	0.23
**Hemoglobin**	136	[121.5;145.0]	129	[119.0;140.0]	0.11
**Erythrocyte count**	4.5	[4.1;4.9]	4.3	[4.0;4.7]	0.08
**Platelet count**	200	[155.0;254.5]	201	[144.0;251.0]	0.4
**Leucocyte count**	5.7	[4.5;7.5]	6.9	[5.4;9.6]	0.081
**Lymphocyte count**	1110.5	[815.8;1528.2]	748.2	[589.6;1184.0]	0.098
**Neutrophil count**	3.8	[2.9;5.5]	5.7	[3.9;8.5]	0.02
**Total protein**	74	[68.2;78.1]	67.9	[65.5;71.5]	0.0055
**Total bilirubin**	9.8	[6.9;13.4]	8.2	[6.6;10.1]	0.15
**Creatinine**	101.5	[84.0;122.0]	101	[90.8;159.2]	0.16
**ALT**	43	[28.0;64.8]	28	[23.8;40.0]	0.0074
**AST**	46	[29.8;64.2]	31	[24.0;52.8]	0.043
**LDH**	327.5	[261.2;409.5]	301.5	[241.8;508.5]	0.46
**hs-CRP**	54.5	[25.9;112.4]	127.2	[58.1;176.3]	0.0077
**Ferritin**	485	[236.5;734.0]	398.5	[291.0;832.8]	0.48
**Triglycerides**	1.4	[1.0;1.8]	1.5	[1.2;1.7]	0.43
**Fibrinogen**	5.9	[4.7;7.0]	6.6	[5.6;7.9]	0.06
**Prothrombin time**	11.4	[10.7;12.2]	11.5	[11.0;13.1]	0.3
**D-dimer**	443	[250.0;870.5]	526.5	[381.8;1376.8]	0.085

**Table 7 ijms-23-07937-t007:** Comparison of laboratory and clinical parameters at admission between patients with and without combined endpoint in patients receiving tocilizumab. Data are presented as median and interquartile range [Q25;Q75]. *p*-values are presented according to the Mann–Whitney test.

	Patients with Tocilizumab, No Endpoint Median	Patients with Tocilizumab, No Endpoint [Q25:Q75]	Patients with Tocilizumab, Endpoint Median	Patients with Tocilizumab, Endpoint [Q25:Q75]	Mann–Whitney *p*-Value
**Respiratory rate**	20	[18.2;22.0]	21.5	[18.5;22.0]	0.43
**Oxygen saturation**	94	[92.0;95.0]	93.8	[89.2;95.8]	0.38
**Temperature**	38.5	[38.0;39.0]	39	[38.6;39.8]	0.036
**Hemoglobin**	133	[118.5;142.0]	130.5	[110.5;139.5]	0.25
**Erythrocyte count**	4.5	[4.1;4.8]	4.2	[3.8;4.7]	0.13
**Platelet count**	210.5	[155.0;256.2]	166	[127.5;237.0]	0.1
**Leucocyte count**	5.8	[4.2;7.8]	6.5	[3.9;7.9]	0.48
**Lymphocyte count**	963.8	[1.4;1250.2]	697.1	[149.9;1056.6]	0.29
**Neutrophil count**	4.3	[2.9;5.9]	4.8	[3.0;7.1]	0.29
**Total protein**	68.7	[65.4;75.8]	67.8	[62.0;71.5]	0.19
**Total bilirubin**	9.1	[6.5;11.4]	7.1	[5.1;10.1]	0.17
**Creatinine**	101.5	[78.0;122.0]	101	[94.2;181.8]	0.12
**ALT**	42	[30.8;65.5]	27.5	[21.5;34.8]	0.0081
**AST**	55	[34.2;71.8]	34	[24.0;52.8]	0.028
**LDH**	370	[292.9;431.0]	301.5	[241.8;508.5]	0.29
**hs-CRP**	89.2	[55.2;128.4]	84.3	[47.4;172.9]	0.46
**Ferritin**	589	[377.0;888.0]	398.5	[309.2;639.0]	0.15
**Triglycerides**	1.5	[1.1;2.0]	1.5	[1.2;1.7]	0.34
**Fibrinogen**	6.7	[5.2;7.6]	6.3	[4.8;6.8]	0.3
**Prothrombin time**	11.6	[10.4;12.2]	11.1	[10.8;13.1]	0.44
**D-dimer**	448	[345.0;831.0]	4.6	[381.8;1095.0]	0.23

**Table 8 ijms-23-07937-t008:** Dynamic changes in cytokine concentrations after the treatment. Data are presented as the median of log2 of the change in the normalized fluorescence intensity values. *p*-values are presented according to the Wilcoxon test.

	All Patients, Median log2FC (N.Fl.)	Wilcoxon Test *p*-Value	Patients without Tocilizumab, Median log2FC (N.Fl.)	Wilcoxon Test *p*-Value	Patients with Tocilizumab, Median log2FC (N.Fl.)	Wilcoxon Test *p*-Value
**IL-6**	0	0.078	−0.27	0.053	1.98	0.00027
**IL-1RA**	−0.36	0.0032	−0.25	0.11	−0.41	0.0046
**IL-10**	−0.42	0.000021	−0.39	0.003	−0.43	0.0016
**G-CSF**	−0.25	0.031	−0.26	0.02	−0.03	0.49
**IL-4**	−0.08	0.072	−0.13	0.085	−0.07	0.33
**IL-13**	−0.07	0.021	−0.06	0.14	−0.1	0.058
**TGF-a**	−0.07	0.042	−0.14	0.0042	−0.01	0.85
**IL-8**	−0.18	0.011	−0.19	0.025	−0.18	0.22
**VEGF**	−0.01	0.072	−0.01	0.083	0	0.59
**MCP-3**	−0.02	0.097	−0.04	0.12	−0.01	0.46
**IL-1a**	−0.2	0.00001	−0.24	0.00018	−0.16	0.015
**MIP-1a**	0.03	0.34	−0.03	0.24	0.05	0.85
**IL-7**	0.01	0.91	−0.04	0.072	0.11	0.058
**IL-17A**	−0.07	0.01	−0.08	0.011	−0.04	0.33
**GM-CSF**	−0.06	0.12	−0.12	0.03	−0.02	0.85
**IL-15**	−0.16	0.00000053	−0.14	0.00035	−0.17	0.0004
**IL-1b**	0	0.8	−0.05	0.084	0.08	0.14
**IFNg**	−0.14	0.0012	−0.21	0.001	0.01	0.48
**TNFb**	−0.05	0.017	−0.03	0.046	−0.08	0.18
**FGF-2**	−0.05	0.025	−0.14	0.0042	−0.02	0.98
**GRO**	−0.11	0.42	−0.08	0.82	−0.2	0.27
**IL-9**	−0.07	0.038	−0.05	0.076	−0.09	0.49
**PDGF-AA**	0.15	0.41	−0.01	0.9	0.41	0.12
**PDGF-AB/BB**	0.25	0.29	0.07	0.96	0.27	0.11
**sCD40L**	−0.03	0.23	−0.12	0.14	0.03	0.99
**IL-12p70**	−0.11	0.00072	−0.07	0.011	−0.17	0.046
**IP-10**	−1.13	0.000001	−1.33	0.00012	−0.44	0.005
**Flt-3L**	−0.03	0.39	−0.03	0.39	−0.03	0.64
**MCP-1**	−0.02	0.81	−0.02	0.34	0.2	0.69
**MDC**	−0.06	0.26	−0.06	0.65	−0.06	0.24
**MIP-1b**	0.35	0.00065	0.38	0.0032	0.26	0.062
**Fractalkine**	−0.07	0.16	−0.07	0.065	−0.06	0.86
**IFNa2**	−0.13	0.00015	−013	0.003	−0.12	0.014
**IL-5**	−0.01	0.64	0.03	0.99	−0.04	0.39
**IL-2**	−0.03	0.035	−0.01	0.062	−0.07	0.34
**TNFa**	0.03	0.75	0.03	0.96	0.05	0.68
**IL-3**	−0.06	0.1	−0.05	0.17	−0.06	0.4
**IL-12p40**	−0.04	0.06	−0.08	0.054	−0.03	0.57
**Eotaxin**	0.3	0.095	0.23	0.89	0.41	0.012
**EGF**	−0.15	0.5	−0.16	0.23	−0.09	0.69

N.Fl.—normalized fluorescence intensity; FC—fluorescence change.

**Table 9 ijms-23-07937-t009:** Criteria for the moderate and severe courses of COVID-19.

Moderate COVID-19	Severe COVID-19
Body temperature 37.5–38.9 °C Respiratory rate 22–29/min; SpO2 94–99%; Grade 1–2 of COVID-19 pneumonia on CT;	Body temperature ≥ 39 °C Respiratory rate ≥ 30/min SpO2 ≤ 93% Grade 3–4 of COVID-19 pneumonia on CT; Decreased level of consciousness

**Table 10 ijms-23-07937-t010:** Clinical characteristics of patients.

Mean age ± SD, years	60.4 ± 14.5
Sex (male), %	49
Coronary heart disease (CHD), %	10
Prior stroke, %	9
Prior myocardial infarction, %	6
Hypertension, %	59
Diabetes mellitus, %	23
Chronic obstructive pulmonary disease (COPD) or asthma, %	6
Chronic kidney disease (CKD), %	18
Anticoagulant therapy, %	100
Steroids, %	30
Tocilizumab, %	53

## Data Availability

The data presented in this study are available in the article and supplementary material. Initial patient data used in this study are available on request from the corresponding author. The data are not publicly available due to the data safety and privacy reasons.

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
