# Peer review of "Comprehensive Cytokine Profiling of Patients with COVID-19 Receiving Tocilizumab Therapy"

_ijms, 2022, doi:10.3390/ijms23147937_

Round 1

Reviewer 1 Report

Review:

The manuscript by Lebedeva et al is a comprehensive analysis of cytokine levels in COVID19 patients receiving Tocilizumab, an antibody that blocks IL-6 function by targeting its receptor. Overall this is a solid and well described study with appropriate references. Of the cytokines analyzed, only IL-1RA levels were prognostically significant. Interestingly, there was little evidence of a COVID19 “cytokine storm syndrome” (as assessed by plasma cytokine levels) with or without Tocilizumab therapy. As the authors note, in COVID-19-related cytokine storm, the levels of cytokines were found to be significantly lower than those observed in the other pathologies. Such work may benefit other therapeutic strategies to treat COVID19. 

Minor suggestions and comments:

11)      In the absence of much prognostic value of individual cytokine levels (other than IL-1RA), the authors in the Discussion should mention the possibility that a cytokine signature, involving threshold levels of many cytokines, might be prognostic. For example could a signature that includes specific cutoff levels of IL-6, IL-1RA, IL-10, and G-CSF hold some prognostic value?

22)      Abstract, page 1: "After a multiplex measurement of 41 circulating cytokines on admission and during the follow-up [...]".  Consider revising this sentence for clarity.

33)      Results 2.2, page 6: "We found(,) that while applied without randomization [...]".   Consider revising this sentence for clarity.

44)      Materials and methods, page 21: "Samples, collected for the standard blood testing, we additionally utilized for a multiplexed assessment of 41 cytokines in the blood of patients using a bead-based assay. " Consider revising this sentence for clarity.

Author Response

Thank you very much for a generally positive evaluation of our work and for your constructive recommendations. Please find below our point-by-point response to the remarks. Also, we attach a revised manuscript file with a marked-up copy of the changes we made to the previous version of the article.

1) In the absence of much prognostic value of individual cytokine levels (other than IL-1RA), the authors in the Discussion should mention the possibility that a cytokine signature, involving threshold levels of many cytokines, might be prognostic. For example could a signature that includes specific cutoff levels of IL-6, IL-1RA, IL-10, and G-CSF hold some prognostic value?

We followed the recommendation and in the amended version of our paper discussed the importance of the cytokine signature for the prognostic evaluation of the COVID-19 patients (page 28 lines 378-381 and page 33 line 480).

2) Abstract, page 1: "After a multiplex measurement of 41 circulating cytokines on admission and during the follow-up [...]". Consider revising this sentence for clarity.

We clarified the sentence according to the recommendation (page 1 lines 21-24).

3) Results 2.2, page 6: "We found(,) that while applied without randomization [...]". Consider revising this sentence for clarity.

We clarified the sentence according to the recommendation (page 7 lines 119-121).

4) Materials and methods, page 21: "Samples, collected for the standard blood testing, we additionally utilized for a multiplexed assessment of 41 cytokines in the blood of patients using a bead-based assay. " Consider revising this sentence for clarity.

We clarified the sentence according to the recommendation (page 31 lines 418-419).

Reviewer 2 Report

This manuscript y Lebedeva et al provides detailed analysis of cytokine profiles in patients with mild and severe COVID-19 disease undergoing therapy with tocilizumab or without tocilizumab therapy and correlated clinical findings on cytokines and other laboratory parameters to clinical outcomes. The study is well done and is certainly timely and important because despite global vaccination efforts, COVID19 remains a major burden on healthcare systems worldwide. The manuscript is written in a clear and logical manner, and conclusions of the study are supported by the presented data. I have only minor concerns of editorial nature that the authors may consider to further improve clarity and presentation of their manuscript.

1.      In the legend to Figure 1, please define “endpoints” and “IL-6 Blocker” labels.

2.      Please describe what you consider Cluster 1 and Cluster 2 either in the legend to Figure 1 or include additional labels for Clusters directly on the figure.

3.      Please define “Time point 1” and “Time point 2” in the manuscript text upon fist mentioning of these terms. What specifically was a time point 1 or 2? This is unclear.

4.      Please re-label Tables to not include any sub-table labeling. Specifically, please label Table 1A as Table 1; Table 1B as Table 2, and so on.  

5.      On the page 6, please replace the word “randomized” in the sentence “…patient subgroups randomized by the initial upregulation…” for the word “stratified”, as no randomization was done but rather a stratification.

Author Response

Thank you very much for a generally positive evaluation of our work and for the constructive remarks and recommendations. Please find below our point-by-point response to the remarks. Also, we attach a revised manuscript file with a marked-up copy of the changes we made to the previous version of the article.

1. In the legend to Figure 1, please define “endpoints” and “IL-6 Blocker” labels.

We added the definition of labels to the legend of Figure 1 (page 5 lines 90-94).

2. Please describe what you consider Cluster 1 and Cluster 2 either in the legend to Figure 1 or include additional labels for Clusters directly on the figure.

We added the definition of Clusters to the legend of Figure 1 (page 3 lines 76-78).

3. Please define “Time point 1” and “Time point 2” in the manuscript text upon fist mentioning of these terms. What specifically was a time point 1 or 2? This is unclear.

We added the definition of Timepoints according to the recommendation (page 5 lines 90-94). The same definition is also mentioned in the “Materials and Methods” section of the Manuscript (page 31 lines 419-422)

4. Please re-label Tables to not include any sub-table labeling. Specifically, please label Table 1A as Table 1; Table 1B as Table 2, and so on.

We relabeled all the Tables according to the recommendation.

5. On the page 6, please replace the word “randomized” in the sentence “…patient subgroups randomized by the initial upregulation…” for the word “stratified”, as no randomization was done but rather a stratification.

We corrected the sentence accordingly (page 7 line 122).